# Dentin Matrix Protein 1 on Titanium Surface Facilitates Osteogenic Differentiation of Stem Cells

**DOI:** 10.3390/molecules26226756

**Published:** 2021-11-09

**Authors:** Suchada Kongkiatkamon, Amsaveni Ramachandran, Kent L. Knoernschild, Stephen D. Campbell, Cortino Sukotjo, Anne George

**Affiliations:** 1Bangkok Hospital Dental Center, Bangkok Hospital, Bangkok 10310, Thailand; 2BDMS Wellness Clinic, Bangkok Dusit Medical Services, Public Company Limited, Bangkok 10330, Thailand; 3Brodie Tooth Development Genetics and Regenerative Medicine Research Laboratory, College of Dentistry, University of Illinois at Chicago, Chicago, IL 60612, USA; aramach@uic.edu (A.R.); anneg@uic.edu (A.G.); 4Department of Restorative Sciences, The Dental College of Georgia, Augusta University, Augusta, GA 30912, USA; kknoernschild@augusta.edu; 5Department of Restorative Dentistry, College of Dentistry, University of Illinois at Chicago, Chicago, IL 60612, USA; stephend@uic.edu (S.D.C.); csukotjo@uic.edu (C.S.); 6Department of Prosthodontics, School of Dental Medicine, Bahçeşehir University, 34353 İstanbul, Turkey

**Keywords:** dental implant, titanium, surface modification, DMP1, stem cell

## Abstract

Dentin matrix protein 1 (DMP1) contains a large number of acidic domains, multiple phosphorylation sites, a functional arginine-glycine-aspartate (RGD) motif, and a DNA binding domain, and has been shown to play essential regulatory function in dentin and bone mineralization. DMP1 could also orchestrate bone matrix formation, but the ability of DMP1 on Ti to human mesenchymal stem cell (hMSC) conversion to osteoblasts has not been studied. There is importance to test if the DMP1 coated Ti surface would promote cell migration and attachment to the metal surface and promote the differentiation of the attached stem cells to an osteogenic lineage. This study aimed to study the human mesenchymal stem cells (hMSCs) attachment and proliferation on DMP1 coated titanium (Ti) disks compared to non-coated disks, and to assess possible osteoblastic differentiation of attached hMSCs. Sixty-eight Ti disks were divided into two groups. Group 1 disks were coated with dentin matrix protein 1 and group 2 disks served as control. Assessment with light microscopy was used to verify hMSC attachment and proliferation. Cell viability was confirmed through fluorescence microscopy and mitochondrial dehydrogenase activity. Real-time polymerase chain reaction analysis was done to study the gene expression. The proliferation assay showed significantly greater cell proliferation with DMP1 coated disks compared to the control group (*p*-value < 0.001). Cell vitality analysis showed a greater density of live cells on DMP1 coated disks compared to the control group. Alkaline phosphatase staining revealed higher enzyme activity on DMP1 coated disks and showed itself to be significantly higher than the control group (*p*-value < 0.001). von Kossa staining revealed higher positive areas for mineralized deposits on DMP1 coated disks than the control group (*p*-value < 0.05). Gene expression analysis confirmed upregulation of runt-related transcription factor 2, osteoprotegerin, osteocalcin, osteopontin, and alkaline phosphatase on DMP1 coated disks (*p*-value < 0.001). The dentin matrix protein promoted the adhesion, proliferation, facilitation differentiation of hMSC, and mineralized matrix formation.

## 1. Introduction

Novel modifications in dental implant surfaces could enhance better osseointegration of dental implants [1,2,3]. Since the inception of current implant designs, modifications to improve osseointegration have led to reportedly [4,5,6] favorable implant survival rates for partially edentulous and edentulous patients. Dental implant osseointegration results through the primary stability as influenced by the bone quantity and quality at the implant site through clinical and biological steps [7,8]. Further modifications of the implant surface to enhance bone growth and maintenance are important for initial healing and ongoing implant stability in function. The rough surface of titanium (Ti) implant affects cell proliferation, differentiation, and cellular response [9,10,11,12]. The surface properties of the Ti implant are an important factor for osseointegration such as surface structure, wettability, chemistry, and charge [10,13,14,15].

Various growth factors and cytokines have been used as signaling molecules to direct the regeneration of the desired tissue [16,17]. Bone morphogenetic protein (BMP) is a molecule that can promote bone formation [18,19,20,21] and induces neovascularization during tissue-engineered large bone defects regeneration also induced angiogenesis [22,23]. BMP induces neovascularization during tissue-engineered large bone defect regeneration and also induced angiogenesis [24]. In addition, dentin matrix protein 1 (DMP1), which is the family of glycoprotein, is an extracellular matrix non-collagenous protein containing an arginine-glycine-aspartate (RGD) motif [25]. This protein contains a large number of acidic domains, multiple phosphorylation sites, a functional arg-gly-asp cell attachment sequence, and a DNA binding domain [26]. DMP1 has shown to play essential regulatory function in bone and dentin mineralization [27]. During osteoblast maturation, phosphorylation of DMP1 takes place and then it is transported to the extracellular matrix and helps in the formation of dentin matrix [28].

DMP1 could also orchestrate bone matrix formation, but the ability of DMP1 on Ti to human mesenchymal stem cell (hMSC) conversion to osteoblasts has not been studied. It is important to test if the DMP1 coated Ti surface would promote cell migration and attachment to the metal surface and promote the differentiation of the attached stem cells to an osteogenic lineage. Therefore, this study aimed to study the proliferation of human mesenchymal stem cell (hMSC) cultured on DMP1 coated Ti surfaces by measuring their attachment, proliferation, and the differentiation effect of hMSCs cultured on DMP1 coated Ti disk by the gene expression pattern of runt-related transcription factor 2 (RUNX2), osteoprotegerin (OPG), osteocalcin (OCN), osteopontin (OPN), and alkaline phosphatase (ALP). Developing such a biologically active protein surface would be useful in applications in dental implantology and in oral and maxillofacial surgery.

## 2. Materials and Methods

### 2.1. Preparation of Ti Disks

Commercially available Ti (Type 2) disks (diameter 15 mm and thickness 3 mm) were milled from Ti rods (American Element, Los Angeles, CA, USA). All titanium disks were consistently subjected to 5 steps of polishing (Buehler Metaserve 3000, Buehler, Germany). First, all disks were polished with 1200 grits carbomide paper followed by sonication for 10 min to ensure the removal of polishing debris. Second, all disks were polished with 9 micron diamond slurry using Ultrapol polishing pads followed by 10 min of sonication in distilled water. Third, all disks were polished with a 3-micron diamond slurry using a Texnet 1000 polishing cloth followed by 10 min of sonication in distilled water. Fourth, all disks were polished with 0.02 micron silica suspension liquid using Chemomet polishing cloths followed by 10 min of sonication. The final step was cleaning all disks with 95% ethanol.

All Ti disks were alumina blasted, cleaned, and oxidized by 50:50 volume of 30% hydrogen peroxide (H_2_O_2_) and sulfuric acid (H_2_SO_4_) for two hours and then rinsed with distilled water and allowed to dry for 1 h at 80 °C.

The Ti samples were divided into two groups; group 1 disks were coated with dentin matrix protein 1 and group 2 disks served as control (Figure 1). Sample size was calculated using the software G*Power [29], and it was obtained a total of 68 (*n* = 68) and 34 in each group. The effect size was taken as 0.8, alpha (*p*-value) 0.05, power of the test 0.9, and allocation ratio (size in each group) = 1.

### 2.2. Surface Coating of Ti Disks with DMP1

Thirty-four disks were used in the experimental group and were placed in 16 well plates. In addition, 100 µL of recombinant Dentin Matrix Protein 1 (rDMP1) (Dr. George Laboratory, Chicago, IL, USA) (1 µg/µL) was added to the Ti disks and placed under the UV light for 24 h. This was taken from reference from the study Ahmad et al. [30], where 100 µL of rDMP1 solution is needed to cover the whole Ti surface without any spillage and 1 mg/mL concentration was used to account for the loss of protein coating in this study. The origin of rDMP1 is *E. coli* (BL 21) colonies. DMP1 is a recombinant protein, and it is expressed in bacteria.

Then, two disks from the experimental group (DMP1 coated Ti surface) and 2 disks from the control group (non-coated Ti surface) were exposed to X-ray photoelectron spectrometer (XPS) analysis. The remaining disks were used for the cell culture study.

### 2.3. Surface Characterization of DMP1 Coated Ti Surface

A total of 4 disks (2 disks from each group) were subject to XPS analysis (Kratos AXIS-165, Kratos Analytical, Ltd., Manchester, UK). The chemical analysis of the DMP1 coated Ti surface and non-coated Ti surface was done using monochromatic XPS. The intensity of each element on the Ti disk surface was identified and graphically recorded in counts per second (cps).

### 2.4. Cell Culture

Commercially available hMSCs (Tulane University, New Orleans, LA, USA) were grown in Dulbecco’s modified Eagle medium (DMEM), and the cells were incubated at 37 °C and 5% CO_2_ until confluence was achieved. The medium was supplemented with 15% fetal bovine serum (FBS), 1% penicillin, and streptomycin. Then, 0.5% trypsin was used to detach the confluent cells and centrifuged, counted, and resuspended in DMEM with 10% FBS. Ti disks were placed into 24 well non-tissue culture treated plates. The first plate contained 5 DMP1 surface coated Ti disks (C1) and 5 non-coated Ti disks (NC1). The second plate contained 5 DMP1 surface coated Ti disks (C2) and 5 non-coated Ti disks (NC2). The third plate contained 7 DMP1 surface coated Ti disks (C3) and 7 non-coated Ti disks (NC3). The fourth plate contained 15 DMP1 surface-coated Ti disks (C4) and 15 non-coated Ti disks (NC4). 20 × 10^3^ cells were carefully plated on top of each disk and incubated at 37 °C and 5% CO_2_ for 3 h (first plate), 24 h (second plate), 3 days (third plate), and 21 days (fourth plate).

### 2.5. Cell Proliferation and Fluorescent Assay

The first (C1-NC1), second (C2-NC2), and third (C3-NC3) sets of experimental specimens were harvested after incubation for 3 h, 24 h, and 3 days, respectively. Cell Titer 96 ^®^_ueous_ One Solution Cell Proliferation Assay (Promega, Madison, WI, USA) was performed to determine cell proliferation. This assay uses MTS tetrazolium, which became a blue formazan product with mitochondrial dehydrogenase activity in viable cells. The absorbance of formazan was determined by a microplate reader at 490 nm. Thus, greater absorbance indicated greater cell metabolism. The measurements were performed three times.

The fluorescent assay was used to observe cells attachment, spreading, and morphology after 3 h, 24 h, and 3 days of seeding the cells. Cells were first fixed in 3.7% formaldehyde, permeabilized with 0.1% Triton X-100, and stained in phosphate-buffered saline (PBS). Actin and nuclei of the cells were stained with ActinGreen 488 ReadyProbes Reagent (Molecular Probes, ThermoFisher Scientific, Waltham, MA, USA) and NucBlue Fixed Cell ReadyProbes Reagent (Molecular Probes, ThermoFisher Scientific), respectively. Cells were imaged with a fully automated inverted microscope (Leica DMI6000 B, Leica Microsystem, Wetzlar, Germany), and postprocessing of the images was performed using LAS AF software (Leica, Wetzlar, Germany).

### 2.6. Alkaline Phosphatase Activity (ALP)

ALP activity was calculated as an indicator of enzymatic activity consistent with bone formation. Ten disks (5 disks from each group) from the fourth plate (C4-NC4) were harvested after 21 days in culture. Then, the cells were thoroughly washed twice with PBS and they were then fixed with 1 mL of ice-cold methanol per well for 10 min and thoroughly washed twice with 1 mL PBS. To determine the possible conversion of hMSCs to osteoblasts, the ALP Conjugate Substrate assay was performed (Bio-Rad, Hercules, CA, USA). In addition, 300 µL of AP reagent A was mixed with 300 µL of AP reagent B (equal amount) and 1 × AP Color Developer Buffer. Then, 1 mL of the mixed reagent was added to each sample and the specimens were incubated for 45 min. Next, the reaction was ended by washing with PBS 3 times and allowing it to air dry. The images were analyzed for an ALP positive area using an image analysis system (ImageJ, Research Services Branch, NIH, Bethesda, MD, USA) and expressed as a percentage by using the formula: [(stained area/total disk area) × 100] (%).

### 2.7. Von Kossa Staining

von Kossa staining was performed to identify the presence of calcium deposits as a possible precursor to bone formation. Ten Ti disks from the fourth plate (5 disks from DMP1 coated group, NC4, and 5 disks from the control group, C4) were harvested after 21 days of incubation. Then, all disks were gently washed with 1 mL PBS two times and fixed with 1 mL of 10% formalin in each well for 15 min. Disks were then thoroughly washed twice with 1 mL deionized water. After air-drying for 20 min, the disks were stained with 1 mL 1% silver nitrate solution for 45 min in the dark. The disks were thoroughly washed 3 times with 1 mL of tap water and 1 mL of a developer was added into each well for 1-5 min. All disks were rinsed with 1 mL of tap water and allowed to air dry. The mineralized nodule area representing phosphate was determined using the formula [(stained area/total disk area) × 100] (%), obtained using a digitized image analysis system (ImageE).

### 2.8. Quantitative Real Time-PCR

Osteogenic differentiation was analyzed by gene expression analysis using a quantitative real-time polymerase chain reaction (qRT-PCR). Total RNA was extracted from cells cultured for 21 days with differentiating medium on the 10 Ti disks (5 disks from each group; DMP1 Ti coated surface (C4) and non-coated surface (NC4) using TRIzol (Invitrogen, Carlsbad, CA, USA) and the purification column [31,32]. The procedure was completed following manufacturer recommendations. Following DNase, I treatment, 25.0 ng (5.0 µL) was taken from each sample and converted to cDNA by using RT^2^ First Strand Kit (SABioscience, Federick, MD, USA). Specific primer sequences (https://www.idtdna.com) were utilized for qRT-PCR (Table 1). The expression osteogenic genes were determined, namely runt-related transcription factor 2 (RUNX2), osteoprotegerin (OPG), osteocalcin (OCN), osteopontin (OPN), and alkaline phosphatase (ALP). Glyceraldehyde 3-phosphate dehydrogenase (GAPDH) was utilized as an internal assay control.

The relative gene expression level was estimated by transforming the logarithmic values into absolute values using the 2−ΔΔCT method, where the average threshold cycle (CT) values were used to quantify the gene expression in each sample: −ΔΔCT = −(ΔCT, Target−ΔCT, GAPDH).

### 2.9. Scanning Electron Microscopy (SEM) Analysis

A total of 4 disks (2 disks from each group, (C4-NC4)) were subject to SEM analysis (STEM, JEM-ARM200CF, JEOL, Inc., Tallahassee, FL, USA) after hMSCs were cultured for 21 days. Effects of surface structure on the cell shape and orientation were analyzed. All specimens were thoroughly washed for 5 min in 1 mL distilled water. The dehydration of SEM samples was performed by rinsing twice for 5 min in 1 mL 30% and 50% ethanol and once for 10 min serially in 70%, 80%, 90%, 95% and 100% ethanol. The final dehydration step was performed in 1 mL hexamethyldisilazane reagent (HMDS, Sigma Aldrich, St. Louis, MO, USA) for 5 min. Specimens were air-dried at room temperature and sputter-coated using an SEM Gold-Coating unit and visualized under the microscope.

### 2.10. Statistical Analysis

Data were entered in Microsoft Excel and analyzed in the SPSS 20 (SPSS Inc., Chicago, IL, USA). Significant differences between different DMP1 coated and non-coated Ti samples were determined using the 2-sample *t*-test. The significant level was set at *p*-value = 0.05.

## 3. Results

### 3.1. DMP1 Coatings on Ti Disks

Figure 2 shows the average intensity of each element on Ti disks with XPS analysis. Control disks (Figure 2A) showed high intensity for Ti (14.35%). O at 530.0 eV is typical for TiO_2_. For DMP1 disks (Figure 2B), the increase in the peak area for N and a decrease in Ti intensity indicated DMP1 presence on the Ti surface after the DMP1 application.

### 3.2. DMP1 Promoted Cell Proliferation

The fluorescence assay indicated the cell’s morphology cultured on control and DMP1 coated disks. All cells were spindle-shaped on both groups, and more cells were noticed on day 21 for DMP1 surfaces compared to control (Figure 3).

The MTS assay of cell mitochondrial activity indicated that cells plated on DMP1 coated and uncoated disks proliferated at the same rate at 3 and 24 h. No statistically significant difference between coated and uncoated Ti disks (*p*-value > 0.05) was observed. However, at day 3, the mean absorbance value for DMP1 coated Ti disk group showed significantly more than the non-coated disk group (*p*-value < 0.001) (Figure 4).

### 3.3. DMP1 Promoted Cell Differentiation and Extracellular Matrix Formation

ALP staining on the 21-day culture revealed significantly greater active enzyme regions between DMP1 coated and uncoated disks (Figure 5). In Figure 5A, the staining reaction was more intense on DMP1 coated Ti disk (16.535% ± 0.1%) and was statistically significantly higher than the non-coated group (0.0025% ± 0.0025%) (*p*-value < 0.001). von Kossa staining showed the cultured hMSCs at day 21 on the DMP1 coated Ti disks had statistically significantly higher calcium mineral deposits (23.261 ± 0.1) than the culture on the non-coated Ti disks (1.71 ± 0.1) (*p*-value < 0.05) (Figure 5B). SEM analysis showed that the DMP1 coated surface promoted hMSCs to synthesize more extracellular matrix (ECM) compared to non-coated at day 21 (Figure 5C).

### 3.4. DMP1 on Coated Disks Increased Osteogenic Gene Expression

The cDNA was checked for quality by PCR with GAPDH Primers. Figure 6A shows the cDNA samples from each sample. Lane A and lane B are cDNA isolated from hMSCs incubated on non-coated titanium surface samples. Lanes C and D are cDNA isolated from hMSCs incubated on DMP1 coated titanium surface samples.

qRT-PCR data indicated that the expressions of OPG, RUNX2, OPN, OCN, and ALP on DMP1 coated Ti specimens were 2.44, 9.94, 13.19, 1.38, and 7.035 fold increase respectively when compared to non-coated Ti surface specimens (*p*-value < 0.001) (Figure 6B).

## 4. Discussion

The importance of endosseous dental implant surface topography has been demonstrate in many studies [33,34,35,36], but until the late 1990s, investigation focused on micron-scale modifications. More recently, the focus has shifted to the nanoscale level. The observation that a micron-scale rough surface prepared by grit blasting and subsequent hydrofluoric acid (HF) treatment presented a superimposed nanotopography suggested that nanoscale modifications could alter adhered cellular activity or tissue responses leading to greater osteogenesis [37,38,39,40].

This study demonstrated that the biomimetic coating of Ti surfaces with DMP1 enhanced the attachment and proliferation of hMSCs. DMP1 contains an integrin binding RGD domain and integrins present on hMSCs can bind to DMP1 coated on Ti disks. Strong cell attachment is necessary for the differentiation and proliferation of hMSCs. In vitro experiments in this study showed that collagen and RGD peptides immobilized on Ti enhanced the adhesion of hMSCs. Results further suggested that DMP1 on Ti surfaces could facilitate hMSC osteogenic conversion toward bone-forming cells.

To confirm osteogenic differentiation of the hMSCs, gene expression analysis was performed after 21 days in culture. Significant changes in this study were observed with RUNX2 expression, which is a master regulatory gene for osteoblastogenesis [41]. This suggests that differentiation of hMSCs to osteoblasts was occurring. Another early osteogenic gene that was significantly upregulated was OPN, whereas ALP, OPG, and OCN appear later with mineralization. It has been shown that variations in the temporal pattern of expression for a variety of markers from various cell culture studies [42].

A lesser magnitude of increase was observed with OCN compared to OPN, OPG, and ALP. As OCN is considered one of the late bone markers [41,43], it can be assumed that OCN will be upregulated after 21 days. Future studies of longer duration could show a significantly greater magnitude in OCN upregulation compared to that observed in this study.

ALP and von Kossa staining assays are indicative of osteogenic differentiation and mineralized matrix deposition. This study confirmed significantly greater enzyme activity and a greater density of mineral deposits on DMP1 disks compared to the controls. Furthermore, SEM analysis at 21 days identified more extracellular matrix formed by hMSCs cultured on the Ti-DMP1 surfaces compared to control. This further supports our hypothesis that DMP1 coated surface may facilitate cellular adhesion and support mineral deposition. Such properties would be beneficial for the osteointegration of dental implants.

To achieve osseointegration, the adherent cells on the Ti surface must differentiate to mineralized matrix producing osteoblasts based on the nano or microtopography of the Ti surface [34,35]. Recently, studies are focusing on the nanoscale level [12,44]. The nano-surface promotes osteogenesis by altering the cellular activity and tissue responses [37,38,39]. Similarly, in our study, DMP1 on the Ti source provided the nanoscale topography for cell viability and differentiation.

This pilot study provides the groundwork for future clinical and translational research regarding the effects of DMP1 on implant osseointegration. For clinical relevance, identification of a means for stable coating of DMP1 to the Ti surface must be achieved. After identifying the coating procedure, appropriate concentrations of DMP1 should be ascertained to attain a high osteogenic response. When these levels are identified, future animal and human studies identifying the influence of this nanostructure modification of Ti disks on short and long-term osseointegration would be possible. In addition, DMP1 may be an effective osseous mediator/promoter in conjunction with bone grafting materials in the maintenance of extraction sockets and the augmentation of edentulous sites. Future studies can explore the releasing rate and mechanism of DMP1 when loaded into Ti nanotubes over time.

From this study, the results showed that attached cells on the Ti and Ti-DMP1 coated surface were vital and healthy, as demonstrated by the green fluorescence emission. Additionally, the number of vital cells attached to the DMP1 coated Ti surface was higher than the non-coated Ti surface. The conclusion that can be drawn from this study is that DMP1 promotes cell proliferation and is nontoxic to hMSCs. Furthermore, the clinical implications of this study relate to the effective use of current technology to heighten the ability of the clinician to predictably achieve osseointegration in the shortest biologically allowable time frame. The inflammatory and healing response after implant placement occurs in parallel. Osteoconductive properties of Ti surface and simultaneous osteoinduction with DMP1 could lead to faster de novo bone formation and implant secondary stability. DMP1 might also contribute to an accelerated healing process and possibly improved bone quality adjacent to the implant.

The current study has some limitations. This in vitro study may not represent the real clinical situation. The current methods to apply DMP1 to titanium surfaces need further improvement. Clinically, DMP1 protein must adhere firmly to the titanium surface because the protein might strip off the surface during placement in the oral cavity. Furthermore, the amount and stability of DMP1 bonded on titanium should also be determined in future studies. If the amount of protein on the titanium surface can be quantitated, then the physiological amount of DMP1 required to trigger cellular activities could be used for coating. Previously, Hamlekhan et al. [45] studied the role of TNT dimensions on drug release over time. They loaded different dimensions of TNTs with a model drug. They found, with the increase of any parameters, the duration of the drug release through a diffusion-limited process. In the future, we are planning to load TNT with DMP1 and study the drug release mechanism. An animal study investigating the effect of Ti-DMP1 on osseointegration is necessary.

## 5. Conclusions

The dentin matrix protein promoted the adhesion and proliferation, and facilitates differentiation of human stem cells and facilitated mineralized matrix formation. Hence, such biologically modified Ti surface with dentin matrix protein may be utilized for the better osseointegration of Ti implants.

## Figures and Tables

**Figure 1 molecules-26-06756-f001:**
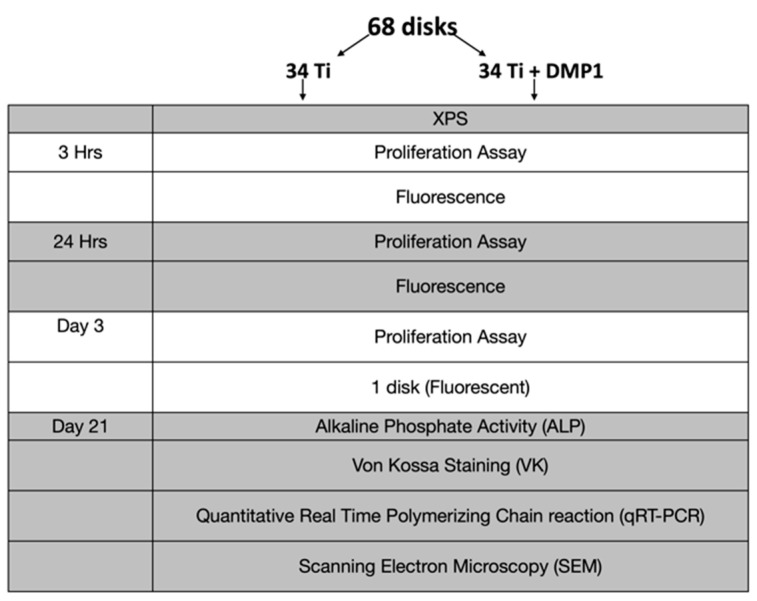
Methodologies summarized flow chart.

**Figure 2 molecules-26-06756-f002:**
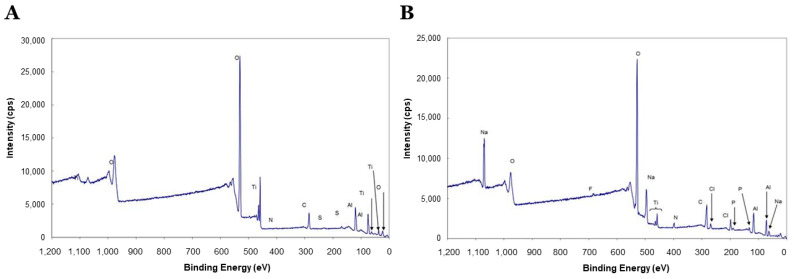
X-ray photoelectron spectrometer (XPS) analysis for uncoated Ti disk (**A**) and DMP1 coated Ti disk (**B**).

**Figure 3 molecules-26-06756-f003:**
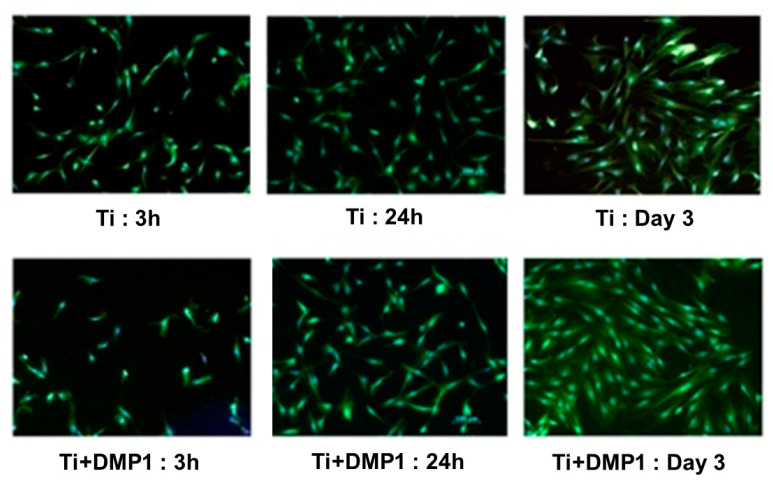
Fluorescence microscopic images of attach cell and cell proliferation assay absorbance for titanium coated and control disks at 3 and 24 h and Day 3.

**Figure 4 molecules-26-06756-f004:**
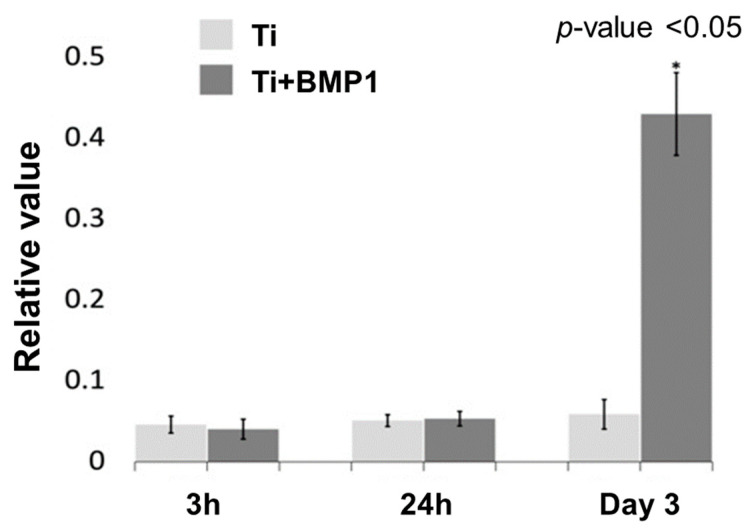
The absorbance value of the (mean ± SD) of attach cell and cell proliferation assay absorbance for titanium coated and control disks at 3 and 24 h and Day 3.

**Figure 5 molecules-26-06756-f005:**
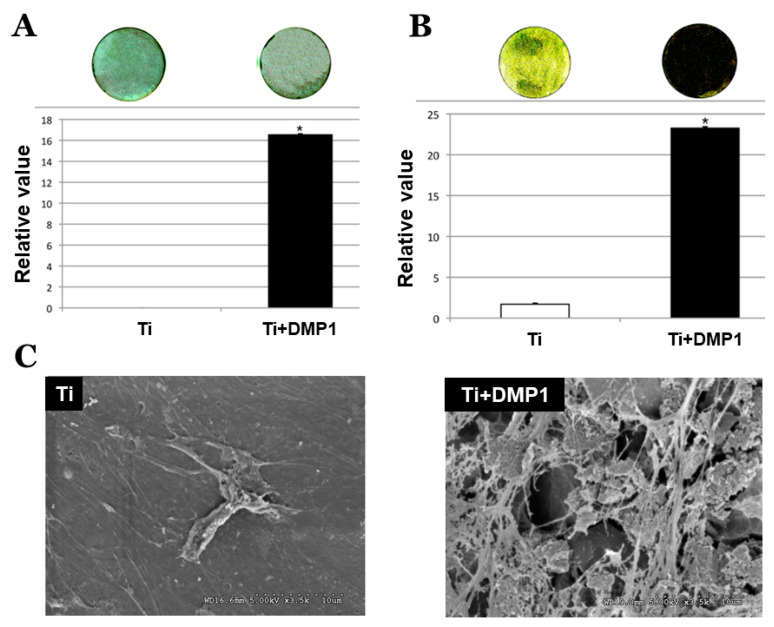
Images of alkaline phosphatase activity and bar graphs represent the means ± SD of triplicate experiments of Ti disk and Ti disk+DMP 1 (**A**). Images of von Kossa staining and bar graph represent the means ± SD of triplicate experiments of Ti disk and Ti disk+DMP 1 (**B**). SEM images of the cellular extracellular matrix on Ti (**C**) and Ti disk + DMP1.

**Figure 6 molecules-26-06756-f006:**
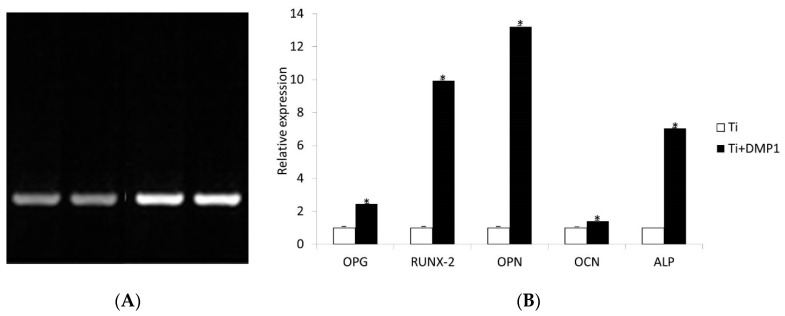
(**A**) Cdna gel of HMSCs for 2 experimental groups (C and NC) (A, B: Non-coated titanium surface and C, D: DMP1 coated titanium surface). (**B**) Quantitative comparison of RT-PCR of OPG, RUNX2, OPN, OCN, and ALP mRNA from hMSCs cultured on DMP1 coated or uncoated disks. All values were normalized to GAPDH mRNA levels. Experiments were performed in triplicate. RT-PCR = Real-time polymerase chain reaction, RUNX2 = runt-related transcription factor 2, OPG = osteoprotegerin, OCN = osteocalcin, OPN = osteopontin, ALP = alkaline phosphatase.

**Table 1 molecules-26-06756-t001:** Various primers used for qRT-PCR in this study.

Gene	Forward (5′–3′)	Reverse (5′–3′)
GADPH	5′-ACAACTTTGGTATCGTGGAAGG-3′	5′-GCCATCACGCCACAGTTTC-3′
RUNX2	5′-TCTCAGATCGTTGAACCTTGCTA-3′	5′-TCTCAGATCGTTGAACCTTGCTA-3′
OPN	5′-AAACCCTGACCCATCTCAGAAGCA-3′	5′-TGGCTGTGAAATTCATGGCTGCTGTGG-3′
OCN	5′-AGCTCAATCCGGACTGT-3′	5′-GGAAGAGGAAAGAAGGGTGC-3′
ALP	5′-ATCGCCTACCAGCTCATGCAT-3′	5′-GTTCAGCTCGTACTGCATGTC-3′
OPG	5′-CAAAGTAATCGCAGAGAGTGTAGA-3′	5′-GAAGGGGAGGTTAGCATGTCC-3′

## Data Availability

The data presented in this study are available on request from the corresponding authors.

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
