# Peer review of "Dentin Matrix Protein 1 on Titanium Surface Facilitates Osteogenic Differentiation of Stem Cells"

_molecules, 2021, doi:10.3390/molecules26226756_

Round 1
Reviewer 1 Report
Dear authors
The paper titled Immobilized Dentin Morphogenic Protein 1 on Titanium Surface Facilitates Osteogenic Differentiation of Stem Cells, presents potential for publication. The study is very interesting; however, some issues/modifications need to be performed for it to be adequate for publication.
Title
Why was the word "Immobilized" in the title?
Introduction and Abstract clearly identify the need and relevance for this subject.
M & M
Methodology targets the main questions appropriately, however some details need to be informed.
Why were 68 Ti disks employed? Was sample size calculation performed?
“100 μl of rDMP1 (1 μg/μl) was added to the Ti disks and placed under the UV light for 24 hrs”. In this paragraph, Is there a standard for placing this amount of rDMP1?
What is the origin of rDMP1?
“Thirty-four disks were used in the experimental group and were placed in 16 well plates”. It was not clear whether It was an album by well?
“Commercially available hMSCs were grown in Dulbecco's modified Eagle medium and the cells were incubated at 37 °C and 5% CO2 until confluence was achieved” What is the origin and passage of the cell?
“medium was supplemented with 15% fetal bovine serum (FBS), 1% penicillin, and strep-tomycin. Then, 0.5% trypsin was used to detach the confluent cells and centrifuged, counted, and resuspended in DMEM with 10% FBS. Ti disks were placed into 24 well non-tissue culture treated plates. The first plate contained 5 DMP1 surface coated Ti disks (C1) and 5 non-coated Ti disks (NC1). The second plate contained 5 DMP1 surface coated Ti disks (C2) and 5 non-coated Ti disks (NC2). The third plate contained 7 DMP1 surface coated Ti disks (C3) and 7 non-coated Ti disks (NC3). The fourth plate contained 15 DMP1 surface-coated Ti disks (C4) and 15 non-coated Ti disks (NC4). 20x 103 cells were carefully plated on top of each Ti disk and incubated at 37°C and 5% CO2 for 3 hours (first plate), 24 hours (second plate), 3 days (third plate), and 21 days (fourth plate).” Why were groups made with different numbers of samples at different evaluation times? This varied number can lead to a bias.
“Cell Titer 96 ®ueous One Solution Cell Proliferation Assay was performed to determine cell attachment and proliferation” Where does Cell Titer 96 ®aqueous One Solution come from? (manufacturer)
“400 μl of prepared VitalDye/DeadDye solution was added to each …” Where does VitalDye/DeadDye solution come from? (manufacturer)
“and 400 μl of Fixation/Hoechst solution was added to each well and the specimens were incubated at room temperature for 10 minutes” Where does Fixation/Hoechst solution come from? (manufacturer)
“ALP activity was calculated as an indicator of enzymatic activity consistent with bone formation. Ten disks (2 disks from each group) from the fourth plate were harvested after 21 days in culture.” Please explain better this division of groups.
It would be more interesting for readers, for the purpose of clarity of understanding, if the authors also presented the methodology summarized in a flowchart.
“Following DNase, I treatment, 25.0 ng (5.0 μl) was taken from each sample and converted to cDNA by using RT2 First Strand Kit.” Where does RT2 First Strand Kit come from? (manufacturer)
“GAPDH was utilized as an internal assay control.” what does GAPDH mean?
Why was the 21-day period not used for adhesion and proliferation and viability testing? I would like these essays to be added as well.
Results
“3.3. DMP1 Promoted Cell Viability and Differentiation” where is the cell viability data?
“HMSCs” please standardize HMSCs or hMSCs
Discussion
It would be interesting for the authors to mention within dentistry, in the context of regeneration, where this DMP1 could be well applied.
Please describe the limitations of this study.
Author Response
Response to Reviewer 1 Comments
Dear authors. The paper titled Immobilized Dentin Morphogenic Protein 1 on Titanium Surface Facilitates Osteogenic Differentiation of Stem Cells, presents potential for publication. The study is very interesting; however, some issues/modifications need to be performed for it to be adequate for publication.
Thank you for your positive comments. Corrections in the Manuscript for Reviewer 1 are highlighted in Green color.
Title
Why was the word "Immobilized" in the title?
Response: Thank you very much for your input. We have modified the title to “Dentin Matrix Protein 1 on Titanium Surface Facilitates Osteogenic Differentiation of Stem Cells”.
Introduction and Abstract clearly identify the need and relevance for this subject.
Response: The relevance of the study is added in the Abstract and Introduction.
M & M
Methodology targets the main questions appropriately, however some details need to be informed.
Why were 68 Ti disks employed? Was sample size calculation performed?
Response: Details on the sample size is added in the MS and explained in the Figure. (Page 3). The sample size was calculated using the software G*Power.
Effect size was taken 0.8, alpha (p value) 0.05, Power of the test 0.9 and allocation ratio (size in each group) = 1.
“100 μl of rDMP1 (1 μg/μl) was added to the Ti disks and placed under the UV light for 24 hrs”. In this paragraph, Is there a standard for placing this amount of rDMP1?
Response: This was taken from reference from the study Ahmad et al. Ref 28.
What is the origin of rDMP1?
Response: The origin of rDMP1 is E. Coli (BL 21) colonies. DMP1 is expressed in bacteria. It is a recombinant protein. (Line 102-104).
“Thirty-four disks were used in the experimental group and were placed in 16 well plates”. It was not clear whether It was an album by well?
Response: We used separate 16 well plated. Details on the sample size is added in the MS and explained in the Figure. (Page 3).
“Commercially available hMSCs were grown in Dulbecco's modified Eagle medium and the cells were incubated at 37 °C and 5% CO2 until confluence was achieved” What is the origin and passage of the cell?
Response: Commercially available HMSCs (Human marrow stem cells purchased from (Tulane University, New Orleans, LA, USA) were grown in the Dulbecco's modified Eagle medium (DMEM). (Line 117-119)
“medium was supplemented with 15% fetal bovine serum (FBS), 1% penicillin, and strep-tomycin. Then, 0.5% trypsin was used to detach the confluent cells and centrifuged, counted, and resuspended in DMEM with 10% FBS. Ti disks were placed into 24 well non-tissue culture treated plates. The first plate contained 5 DMP1 surface coated Ti disks (C1) and 5 non-coated Ti disks (NC1). The second plate contained 5 DMP1 surface coated Ti disks (C2) and 5 non-coated Ti disks (NC2). The third plate contained 7 DMP1 surface coated Ti disks (C3) and 7 non-coated Ti disks (NC3). The fourth plate contained 15 DMP1 surface-coated Ti disks (C4) and 15 non-coated Ti disks (NC4). 20x 103 cells were carefully plated on top of each Ti disk and incubated at 37°C and 5% CO2 for 3 hours (first plate), 24 hours (second plate), 3 days (third plate), and 21 days (fourth plate).” Why were groups made with different numbers of samples at different evaluation times? This varied number can lead to a bias.
Response: English is improved in throughout the manuscript.
“Cell Titer 96 ®ueous One Solution Cell Proliferation Assay was performed to determine cell attachment and proliferation” Where does Cell Titer 96 ®aqueous One Solution come from? (manufacturer)
Response: Cell Titer 96 ®ueous One Solution Cell Proliferation Assay is bought from Promega, Madison, WI, USA. (Line 138)
“400 μl of prepared VitalDye/DeadDye solution was added to each …” Where does VitalDye/DeadDye solution come from? (manufacturer)
Response: Thank you very much for your input. We have modified the manuscript and delete this information.
“and 400 μl of Fixation/Hoechst solution was added to each well and the specimens were incubated at room temperature for 10 minutes” Where does Fixation/Hoechst solution come from? (manufacturer)
Response: Thank you very much for your input. We have modified the manuscript and delete this information.
“ALP activity was calculated as an indicator of enzymatic activity consistent with bone formation. Ten disks (2 disks from each group) from the fourth plate were harvested after 21 days in culture.” Please explain better this division of groups.
It would be more interesting for readers, for the purpose of clarity of understanding, if the authors also presented the methodology summarized in a flowchart.
Response: Details on the sample size calculation is attached below. And the details are mentioned in the MS and the methodology is explained in the Figure 1. (Page 3).
- 15 disks from C (Control), 15 disks from NC were utilized to test
- 5 disks from each group were tested for
- ALP Assay
- Von Kossa Staning
- Gene expression analysis
“Following DNase, I treatment, 25.0 ng (5.0 μl) was taken from each sample and converted to cDNA by using RT2 First Strand Kit.” Where does RT2 First Strand Kit come from? (manufacturer)
Response: RT2 First Strand Kit was bought from SABioscience, Federick, MD, USA. (Line 187-189)
“GAPDH was utilized as an internal assay control.” what does GAPDH mean?
Response: GAPDH = Glyceraldehyde 3-phosphate dehydrogenase. The full form is added. (Line 192)
Why was the 21-day period not used for adhesion and proliferation and viability testing? I would like these essays to be added as well.
Response: Cells attachment is the first step required before cells proliferate and differentiate. We usually perform cells proliferation assay during the early stages (up to 3-7 days). Cells may be confluent and differentiate by 21 days.
Results
“3.3. DMP1 Promoted Cell Viability and Differentiation” where is the cell viability data?
Response: We apologize for the confusion. We have deleted this section.
“HMSCs” please standardize HMSCs or hMSCs
Response: Corrected throughout the manuscript as hMSCs.
Discussion
It would be interesting for the authors to mention within dentistry, in the context of regeneration, where this DMP1 could be well applied.
Response: The importance of DMP1 in dentistry is added in the discussion. (Line 273-279)
Please describe the limitations of this study.
Response: Limitation is added at the end of the discussion. (Line 335-347)

Reviewer 2 Report
In this manuscript, the authors studied the effects of dentin matrix protein 1 (DMP1) coated titanium (Ti) disks on the attachment, proliferation, viability and osteogenic differentiation of human mesenchymal stem cells (hMSCs). They found that DMP1 coated Ti disks promoted cell proliferation and viability, increased alkaline phosphatase activity and von Kossa staining area, and upregulated the expressions of runt-related transcription factor 2, osteoprotegerin, osteocalcin, osteopontin and alkaline phosphatase, in comparison to non-coated control Ti disks. Therefore, the authors conclude that DMP1 promotes adhesion, proliferation and osteogenic differentiation of hMSCs.
Overall, the findings are interesting but the manuscript was not well-written. Here are a number of concerns need to be addressed:
1) Title: Immobilized Dentin Morphogenic Protein 1 on Titanium Surface Facilitates Osteogenic Differentiation of Stem Cells
Comment: “Dentin morphogenic protein 1” should be “dentin matrix protein 1”?
2). Abstract: “Dentin matrix protein 1 (DMP1) is an arginine-glycine-aspartate (RGD) motif and has shown to play essential regulatory function in dentin and bone mineralization.”
Comment: Dentin matrix protein 1 (DMP1) is only “an arginine-glycine-aspartate (RGD) motif”?
3) Introduction: “Bone morphogenetic protein (BMP) is a molecule that can promote bone formation [17-20] and defects [21,22].”
Comment: BMP promotes bone formation and defects? Please specify what defects BMP promotes.
4) Materials and Methods:
a) “Commercially available 68 Ti (Type 2) disks (diameter 15 mm and thickness 3 mm) were milled from Ti rods.”
Comment: The source of Ti rods should be stated.
b) “100 μl of rDMP1 (1 μg/μl) was added to the Ti disks and placed under the UV light for 24 hrs.”
Comment: What does rDMP1 stand for? The source of rDMP1 should be stated. Why did the author choose the amount of rDMP indicated – 100 μl of rDMP1 (1 μg/μl) – in their studies?
c) “Commercially available hMSCs were grown in Dulbecco's modified Eagle medium and the cells were incubated at 37 °C and 5% CO2 until confluence was achieved.”
Comment: The source of hMSCs should be given.
d) "The fourth plate contained 15 DMP1 surface-coated Ti disks (C4) and 15 non-coated Ti disks (NC4). 20x 103 cells were carefully plated on top of each Ti disk and incubated at 37°C and 5% CO2 for 3 hours (first plate), 24 hours (second plate), 3 days (third plate), and 21 days (fourth plate).”
“ALP activity was calculated as an indicator of enzymatic activity consistent with bone formation. Ten disks (2 disks from each group) from the fourth plate were harvested after 21 days in culture.”
“von Kossa staining was performed to identify the presence of calcium deposits as a possible precursor to bone formation. Ten Ti disks from the fourth plate (5 disks from DMP1 coated group and 5 disks from the control group) were harvested after 21 days of incubation.”
“Osteogenic differentiation was analyzed by gene expression analysis using reverse-transcriptase polymerase chain reaction (RT-PCR). Total RNA was extracted from cells cultured for 21 days with differentiating medium on the 10 Ti disks (5 disks from each group; DMP1 Ti coated surface (C) and non-coated surface (NC) using TRIzol (Invitrogen, Carlsbad, CA, USA) and the purification column [26,27].”
“A total of 4 disks (2 disks from each group) were subject to SEM analysis after hMSCs were cultured for 21 days.”
Comment: The authors had a total 30 Ti disks in the fourth plate, and incubated them for 21 days. The authors then used 10 disks for ALP activity assay, 10 disks for von Kossa staining, 10 disks for RNA extraction and real-time PCR and 4 disks for SEM analysis. Therefore, the authors began with 30 Ti disks, but actually used 34 disks for various types of experiments. Please clarify where the extra 4 disks came from.
e) “Osteogenic differentiation was analyzed by gene expression analysis using reverse-transcriptase polymerase chain reaction (RT-PCR).”
Comment: Did the authors really use “reverse-transcriptase polymerase chain reaction (RT-PCR)” to quantify gene expression?
f) The authors should provide the primer sequences used in this study, including the primers for GAPDH, runt-related transcription factor 2 (RUNX2), osteoprotegerin (OPG), osteocalcin (OCN), osteopontin (OPN), and alkaline phosphatase (ALP).
g) “Then, the cells were thoroughly washed twice with PBS and they were then fixed with 1 ml of ice-cold methanol per well for 10 minutes and thoroughly thoroughly washed twice with 1 ml PBS.”
Comment: “thoroughly thoroughly”?
h) “Significant differences between different DMP1 coated and non-coated Ti samples were determined using using the 2-sample T-test. The significant level was set at P value = 0.05.”
Comment: “using using”?
5) Results:
a) Figure 2A and 2B should be combined.
b) “Light microscopic images showed a qualitative increase in the density of live cells on DMP1 coated Ti disks (Figure 3A) when compared with non-coated Ti disks (Figure 3B).”
Figure 3 legend: “Light microscopic images of attached cells on uncoated Ti disk (A) and DMP1 coated disk (B).”
Comment: The description in the text is not consistent with the description in the figure legend. In addition, in the Materials and Methods part, the authors stated “The presence of live or dead cells was observed using fluorescence microscopy.” However, it is not clear which one is a dead cell and which one is a live cell based on the images provided (Figure 3A and 3B). What is the ratio of dead cells to live cells in each group?
c) “SEM analysis showed that the DMP1 coated surface promoted hMSCs to synthesize an extracellular matrix (ECM) that was calcified (Figure 3I, J).”
Comment: It is not clear where “an extracellular matrix (ECM)” is or where is “calcified” based on the images provided (Figure 3I and J). How can SEM confirm mineral deposition?
d) “RT-PCR analysis for the DMP1 coated titanium surface group and non-coated group showed upregulated expression of the tested osteoblastic gene.”
Figure 5 legend: “RT-PCR = reverse-transcriptase polymerase chain reaction,”
Comment: The authors defined “RT-PCR” as “reverse-transcriptase polymerase chain reaction.” Please clarify how to use RT-PCR to quantify gene expression.
Author Response
Response to Reviewer 2 Comments
Thank you for your positive comments. Corrections in the Manuscript for Reviewer 2 are highlighted in Yellow color.
In this manuscript, the authors studied the effects of dentin matrix protein 1 (DMP1) coated titanium (Ti) disks on the attachment, proliferation, viability and osteogenic differentiation of human mesenchymal stem cells (hMSCs). They found that DMP1 coated Ti disks promoted cell proliferation and viability, increased alkaline phosphatase activity and von Kossa staining area, and upregulated the expressions of runt-related transcription factor 2, osteoprotegerin, osteocalcin, osteopontin and alkaline phosphatase, in comparison to non-coated control Ti disks. Therefore, the authors conclude that DMP1 promotes adhesion, proliferation and osteogenic differentiation of hMSCs.
Overall, the findings are interesting but the manuscript was not well-written. Here are a number of concerns need to be addressed:
1) Title: Immobilized Dentin Morphogenic Protein 1 on Titanium Surface Facilitates Osteogenic Differentiation of Stem Cells
Comment: “Dentin morphogenic protein 1” should be “dentin matrix protein 1”?
Response: We have corrected “Dentin morphogenic protein 1” to “Dentin matrix protein 1”.
2). Abstract: “Dentin matrix protein 1 (DMP1) is an arginine-glycine-aspartate (RGD) motif and has shown to play essential regulatory function in dentin and bone mineralization.”
Comment: Dentin matrix protein 1 (DMP1) is only “an arginine-glycine-aspartate (RGD) motif”?
Response: The protein contains a large number of acidic domains, multiple phosphorylation sites, a functional arg-gly-asp cell attachment sequence, and a DNA binding domain. (Line 60-62, Ref 26).
Ref 26. Srinivasan R. et al. Recombinant expression and characterization of dentin matrix protein 1. Connect Tissue Res, 1999. 40(4): p. 251-8.
3) Introduction: “Bone morphogenetic protein (BMP) is a molecule that can promote bone formation [17-20] and defects [21,22].”
Comment: BMP promotes bone formation and defects? Please specify what defects BMP promotes.
Response: BMP induces neovascularization during tissue-engineered large bone defect regeneration, also induced angiogenesis. (Line 56-58, Ref 24).
Ref 24. Pearson HB et al. Effects of Bone Morphogenetic Protein-2 on Neovascularization During Large Bone Defect Regeneration. Tissue Engineering Part A, 2019. 25(23-24).
4) Materials and Methods:
- a) “Commercially available 68 Ti (Type 2) disks (diameter 15 mm and thickness 3 mm) were milled from Ti rods.”
Comment: The source of Ti rods should be stated.
Response: Ti rods were purchased from American Element, Los Angeles, CA, USA. (Line 80)
- b) “100 μl of rDMP1 (1 μg/μl) was added to the Ti disks and placed under the UV light for 24 hrs.”
Comment: What does rDMP1 stand for? The source of rDMP1 should be stated. Why did the author choose the amount of rDMP indicated – 100 μl of rDMP1 (1 μg/μl) – in their studies?
Response: rDMP1 stand for recombinant Dentin Matrix Protein 1 (Line 100)
The source of rDMP1 is added. (Line 100-102)
We followed the study Ahmad et al. where 100 µl of rDMP1 solution was just enough to cover the whole titanium surface without any spillage. 1mg/ml concentration was used to account for loss of protein coating. (Line 102-104, Ref 30).
Ref 30. Ahmad AR et al. Recombinant human dentin matrix protein 1 (DMP1) induces the osteogenic differentiation of human periodontal ligament cells. Biotechnol Rep (Amst). 2019 Sep; 23: e00348.
- c) “Commercially available hMSCs were grown in Dulbecco's modified Eagle medium and the cells were incubated at 37 °C and 5% CO2 until confluence was achieved.”
Comment: The source of hMSCs should be given.
Response: The source of hMSCs is added. (Tulane University, New Orleans, LA, USA) (Line 111)
- d) "The fourth plate contained 15 DMP1 surface-coated Ti disks (C4) and 15 non-coated Ti disks (NC4). 20x 103cells were carefully plated on top of each Ti disk and incubated at 37°C and 5% CO2for 3 hours (first plate), 24 hours (second plate), 3 days (third plate), and 21 days (fourth plate).”
“ALP activity was calculated as an indicator of enzymatic activity consistent with bone formation. Ten disks (2 disks from each group) from the fourth plate were harvested after 21 days in culture.”
“von Kossa staining was performed to identify the presence of calcium deposits as a possible precursor to bone formation. Ten Ti disks from the fourth plate (5 disks from DMP1 coated group and 5 disks from the control group) were harvested after 21 days of incubation.”
“Osteogenic differentiation was analyzed by gene expression analysis using reverse-transcriptase polymerase chain reaction (RT-PCR). Total RNA was extracted from cells cultured for 21 days with differentiating medium on the 10 Ti disks (5 disks from each group; DMP1 Ti coated surface (C) and non-coated surface (NC) using TRIzol (Invitrogen, Carlsbad, CA, USA) and the purification column [26,27].”
“A total of 4 disks (2 disks from each group) were subject to SEM analysis after hMSCs were cultured for 21 days.”
Comment: The authors had a total 30 Ti disks in the fourth plate, and incubated them for 21 days. The authors then used 10 disks for ALP activity assay, 10 disks for von Kossa staining, 10 disks for RNA extraction and real-time PCR and 4 disks for SEM analysis. Therefore, the authors began with 30 Ti disks, but actually used 34 disks for various types of experiments. Please clarify where the extra 4 disks came from.
Response: We have added the correct number in the manuscript.
Details on the sample size calculation is attached below. And the detail is mentioned in the MS and the methodology is explained in the Figure 1. (Page 3).
- 15 disks from C (Control), 15 disks from NC were utilized to test
- 5 disks from each group were tested for
- ALP Assay
- Von Kossa Staning
- Gene expression analysis
Also, this picture below showed how we calculate the sample size.
Effect size was taken 0.8, alpha (p value) 0.05, Power of the test 0.9 and allocation ratio (size in each group) = 1.
- e) “Osteogenic differentiation was analyzed by gene expression analysis using reverse-transcriptase polymerase chain reaction (RT-PCR).”
Comment: Did the authors really use “reverse-transcriptase polymerase chain reaction (RT-PCR)” to quantify gene expression?
Response: I apologize for the confusion. We used qRT-PCR, not RT-PCR. We have modified the text accordingly.
- f) The authors should provide the primer sequences used in this study, including the primers for GAPDH, runt-related transcription factor 2 (RUNX2), osteoprotegerin (OPG), osteocalcin (OCN), osteopontin (OPN), and alkaline phosphatase (ALP).
Response: The primer sequences used in this study is added. (Table 1)
- g) “Then, the cells were thoroughly washed twice with PBS and they were then fixed with 1 ml of ice-cold methanol per well for 10 minutes and thoroughly thoroughly washed twice with 1 ml PBS.”
Comment: “thoroughly thoroughly”?
Response: It is corrected.
- h) “Significant differences between different DMP1 coated and non-coated Ti samples were determined using using the 2-sample T-test. The significant level was set at P value = 0.05.”
Comment: “using using”?
Response: It is corrected.
5) Results:
- a) Figure 2A and 2B should be combined.
Response: Figures 2A and 2B are combined.
- b) “Light microscopic images showed a qualitative increase in the density of live cells on DMP1 coated Ti disks (Figure 3A) when compared with non-coated Ti disks (Figure 3B).”
Figure 3 legend: “Light microscopic images of attached cells on uncoated Ti disk (A) and DMP1 coated disk (B).”
Comment: The description in the text is not consistent with the description in the figure legend. In addition, in the Materials and Methods part, the authors stated “The presence of live or dead cells was observed using fluorescence microscopy.” However, it is not clear which one is a dead cell and which one is a live cell based on the images provided (Figure 3A and 3B). What is the ratio of dead cells to live cells in each group?
Response: Apology for this confusion. We have deleted the live and dead cells assay.
- c) “SEM analysis showed that the DMP1 coated surface promoted hMSCs to synthesize an extracellular matrix (ECM) that was calcified (Figure 3I, J).”
Comment: It is not clear where “an extracellular matrix (ECM)” is or where is “calcified” based on the images provided (Figure 3I and J). How can SEM confirm mineral deposition?
Response: Thank you very much for your input. We have modified the legend and the text in the result and discussion section accordingly. SEM can only confirm mineral deposition if we did an EDX which we have not. However, PCR data is able to show differentiation of the stem cells.
- d) “RT-PCR analysis for the DMP1 coated titanium surface group and non-coated group showed upregulated expression of the tested osteoblastic gene.”
Figure 5 legend: “RT-PCR = reverse-transcriptase polymerase chain reaction,”
Comment: The authors defined “RT-PCR” as “reverse-transcriptase polymerase chain reaction.” Please clarify how to use RT-PCR to quantify gene expression.
Response: I apologize for the confusion. We used QRT-PCR, not RT-PCR. The description on how to use QRT-PCR has been described in the material and method.

Round 2
Reviewer 1 Report
The authors improved the quality of the manuscript after the reviewer's indications. Congratulations!
Reviewer 2 Report
The concerns have been properly addressed in this revised version.